# Collective Bargaining in the Information Economy Can Address AI-Driven Power Concentration

**Nicholas Vincent**
Simon Fraser University
Burnaby, BC, Canada
nvincent@sfu.ca

**Matthew Prewitt**
RadicalxChange
Oakland, CA, USA
matt@radicalxchange.org

**Hanlin Li**
University of Texas at Austin
Austin, TX, USA
lihanlin@utexas.edu

## Abstract

This position paper argues that there is an urgent need to restructure markets for the information that goes into AI systems. Specifically, producers of information goods (such as journalists, researchers, and creative professionals) need to be able to collectively bargain with AI product builders in order to receive reasonable terms and a sustainable return on the informational value they contribute. We argue that without increased market coordination or collective bargaining on the side of these primary information producers, AI will exacerbate a large-scale "information market failure" that will lead not only to undesirable concentration of capital, but also to a potential "ecological collapse" in the informational commons. On the other hand, collective bargaining in the information economy can create market frictions and aligned incentives necessary for a pro-social, sustainable AI future. We provide concrete actions to support a coalition-based approach to achieve this goal. For example, researchers and developers can establish technical mechanisms such as federated data management tools and explainable data value estimation techniques to inform and facilitate collective bargaining in the information economy. Additionally, regulatory and policy interventions may be introduced to support trusted data intermediary organizations representing guilds or syndicates of information producers.

## 1 Introduction

### 1.1 Core position

**In this paper, we argue that members of academic, civic, and industry organizations should support "collective bargaining in the information economy" (CBI) as a means to mitigate AI-driven power concentration and other risks, including widespread market failures and collapse of the informational commons. We propose specific actions to support this agenda.**

Our argument is based on well-documented problems with markets for information (e.g., Arrow's "information paradox" [7]), and the potential for AI progress to exacerbate them to the detriment of the economy as a whole. More specifically, because information can be copied at trivial cost (information is hard to "exclude"), the "price" of information goods in a (perfectly) competitive marketplace approaches zero. This means that actors with powerful means to aggregate, process, distill, and control the flow of information are in a position to capture a disproportionate share of the

39th Conference on Neural Information Processing Systems (NeurIPS 2025) Position Paper Track.

social value of information, leaving all other market actors with very weak incentives to invest in the production of information and knowledge. We believe that collective bargaining in the information economy – covering many types of information, from novel scientific data to user records from LLM products – can preserve or restore critical incentives for the production of valuable new information goods, while mitigating power concentration.

To establish healthy markets for information in the post-AI age, we must develop technical, legal, and social capabilities for collective bargaining over the permissible uses of large and diffuse pools of aggregated information. By this, we mean negotiation between (a) aligned or "trusted" data intermediary organizations representing large groups of information producers [31], and (b) information-aggregating AI operating organizations (primarily, tech companies and certain state actors). The subject of the negotiations would be various terms of data use, including constraints on downstream applications, reporting requirements, future compensation for data contributors, and more.

To further concretize the article's proposal, specific actions that support collective bargaining in the information economy would include:

- **Private Initiatives by Information Producers**: Leaders and workers in information-producing industries (research, journalism, creative professionals, etc.) should begin forming industry- or sector-wide joint ventures to increase their collective bargaining power over rights to train/build AI using their information. Consumers and communities (social, geographic, religious, etc.) should do the same – collecting and controlling the information they generate, and delegating power to trusted representatives to bargain with tech/AI over the rights to use it. Relatedly, these information-producing actors should, to the maximum feasible extent, seek to disseminate and license information in ways that prevent AI companies from training on it through backdoor access instead of negotiating for rights.

- **Public Support by Governments**: Government departments and agencies (e.g., the Federal Trade Commission and Department of Justice in the US and global equivalents like the Canadian Competition Bureau and European Commission's Directorate-General for Competition) should issue clarifying statements, and/or new "safe harbor" regulations clarifying and/or expanding the extent to which multi-stakeholder joint ventures that increase information producers' bargaining power do not violate antitrust[1], data protection, and other rules.

- **Refocused Advocacy and Research**: AI safety advocates should, in concert with information producers and their representatives, invest in advocacy for collective bargaining by information producers, because this approach will reduce the concentration of power in a few AI operators, mitigate foreseeable social and economic risks, and provide a safer path forward for powerful AI. The ML/AI research community should continue to develop techniques to assess the causal impact of data on models, e.g., data valuation, influence, and attribution [39]. These mechanisms will equip both information producers and AI companies with the means to better estimate the true value of new information and compensate it at a socially efficient level. Human-computer interaction (HCI) and design communities should identify effective interaction patterns for individuals making decisions about data pooling, and for trusted representatives participating in bargaining. ML/AI, HCI, and other fields should work together to ensure that state-of-the-art techniques in data value estimation can be brought to bear on practical systems that support informed CBI.

- **Support from Powerful Technology Actors**: Large technology companies should embrace this agenda because (a) CBI can help to ensure that their products use diverse, high-quality data (making their services safer and more reliable) and (b) it will avoid economic distortions that could impoverish their customers and destabilize their market share. Small AI operators should focus on securing high-quality information tailored to their business needs and develop relationships with specific trusted data intermediaries to that end.

## 1.2 Motivation

Market incentives to produce new information systematically fail due to inherent properties of information. In efficient and competitive markets, sellers offer to sell goods and services at their

---

[1]Notably, actions by US states may also be able to expand possibilities for *CBI* using Parker Immunity [10].

marginal unit cost of production; but when information is the good being sold, that marginal cost approaches zero, since information can be easily copied. [7, 19, 49]. Because efficient and competitive markets push the price of information to zero, market actors who invest in producing new information must also control and limit its dissemination – thus reducing the market's competitiveness and/or efficiency – in order to earn *any* return on their investment, let alone one commensurate with the information's diffuse social value (see e.g., discussion of related issues in recent economics and computer science scholarship [28, 11].)

Over centuries of technological history, this troubling paradox with information markets has been repeatedly noticed and addressed through ad hoc interventions aimed at reducing the competitiveness and/or efficiency of information markets. These include passing and enforcing intellectual property rules, preserving secrecy, making public investments in information goods, and establishing private monopolies or market power [12].

Sometimes, non-zero dissemination costs have played an important role in maintaining non-zero prices for information. But in historical periods when advances in technology have removed such "accidental" friction, institutional or legal interventions have been seen as necessary to preserve incentives to produce information. For example, throughout the 18th to 20th centuries publishers, authors, inventors, and policymakers frequently saw a need to strengthen artificial information monopolies – intellectual property rights – to preserve incentives to produce new information goods in the face of plummeting dissemination costs brought about by print, broadcast, and digital technologies [42, 32].

Governments have similarly seen a consistent need to directly intervene in otherwise competitive and efficient information markets, in order to create sufficient incentives for information production. In the Cold War, US markets' insufficient incentivization of basic scientific research inspired massive, non-market-driven government investments in order to keep pace with the Soviet Union scientifically, militarily, and economically. [7, 38, 44]. In the internet era, US courts created new incentives for innovation by dramatically relaxing the enforcement of anti-monopoly rules, thus rewarding the developers of new technology with private ownership of critical infrastructural communications networks. See, for example, Verizon Communications Inc. v. Law Offices of Curtis V. Trinko, LLP, 540 US 398 (2004), in which Justice Scalia explained that heavy antitrust intervention in network industries could "diminish the incentive for the monopolist, the rival, or both to invest in . . . economically beneficial facilities development."

This accumulation of ad hoc fixes to information markets' failures led to a flawed, but provisionally stable status quo in the information economy of the late 20th and early 21st centuries. Cracks arguably emerged about a decade into the 21st century when computational and networking technologies (file sharing, social media networks, etc.) greatly decreased the incentives to invest in socially valuable information such as art and journalism, straining the socio-cultural fabric. Now, "AI" technologies have radically disrupted that already-shaky ecosystem of incentives, through what can be understood in part as combined massive advances in the ability to compress [13], distill [1], and interpolate information in minute detail [59]. This has implications for every kind of informational good (e.g., "knowledge" [53], "data" [4], and "content" [43]). We detail the existing and potential changes in incentives around information in Section 2.2.

The upending-by-AI of incentives in the information economy forces society to make a difficult choice either by commission or omission. On the one hand, if failed information markets proliferate without correction by states or other public-interest actors, we argue that the social value of most recorded information is likely to accrue over time to highly concentrated pools of political or financial capital. The most powerful actors in society will be able to profit from information uniquely, because they will have the scale and access to capital needed to exploit and/or control new information (such as by manufacturing new inventions while keeping some information private). These actors will then supply the friction and structure that information markets need to incentivize new information goods, but they will do so without plausible accountability to public interests.

Achieving monopoly-like market power through concentration of political and/or financial capital will emerge as the clearest pathway to "circumventing" broken information markets, accelerating a rush to leverage new information (and/or manipulate the informational commons through propaganda), for financial or political power. By contrast, weaker actors in competitive markets will not be able to profit from information. Consequently, distributed and disinterested contributions to the informational commons will be disincentivized. There is no natural limit to the degree of market

power and concentration that could emerge, since every combination between market actors will be "synergistic", i.e., more powerful than the sum of its parts. Taken to an extreme, this could result in what we term a "capital singularity": concentration of financial, informational, and market power into very few actors or even a single actor.

More optimistically, society might embark upon the project of establishing an intentionally less-concentrated market structure, with multiple sites of durable market power, mediating the rights to use information in AI products. If successful, this will forestall the extreme concentration of power, and preserve diverse incentives for the general population to participate in advancing knowledge and human creativity, and the social fabric that depends upon those projects. Moreover, it will not impede information technology's progress but enhance it, by ensuring AI systems remain supplied with a constant stream of human-produced information with a diversity of motives – in other words, information reflecting not just the imperatives of monopolists, but the diverse products of intentional human creation.

Accordingly, we believe that *CBI* is the shortest (and perhaps only) path to fulfill the latter vision in which human creation collectively powers more useful models and creates a more sustainable AI ecosystem.

In arguing for *CBI*, we draw on scholarship on "data as labor" [6, 41], "data leverage" [52], "data dignity", and "plurality" [56]. A core idea in this body of work is that much of the value provided by AI products can be traced to "data labor activities" [33] performed by people. Companies acquire valuable data by exploiting their bargaining power in billions of seemingly low-stakes data-for-service transactions with individuals. The work mentioned above and other relevant studies have explored ways to communicate data's value to the public (e.g., [54]), support collective action with data [51], create new data market structures or revenue-sharing approaches [4], and so on. For instance, researchers have begun to directly study data-sharing consortium approaches with promising early results [11].

We also draw on the information economics literature. Classical work has highlighted the challenging economics of interrelated concepts such as innovation [7] and knowledge [53]. More recent empirical work studying the economic characteristics of information and data [28, 4] has highlighted how early predictions about inefficiencies and market failures for information have held up in light of computing progress. We also look at analyses of collective action [40, 51] and collective bargaining [16] for insight into the factors that are likely to produce successful coordination. We expect that the practical implementation of CBI will require looking to existing studies on existing initiatives around data cooperatives [20] and gig worker organizing [60, 47]. Finally, our arguments are inspired by historical precedents that involved technological disruption to informational rights [12].

## 1.3 Target Audience

By highlighting the need for *CBI*, we aim to coordinate initiatives from across many domains, so that (1) technical and policy interventions can complement each other (e.g., new tools for collective bargaining can be built in conjunction with legal and regulatory support for the bargaining intermediary organizations) and (2) efforts to support collective bargaining can be designed with other, parallel interventions in mind (e.g., a push towards openness [15, 9] in the AI space and/or "public AI" [26]).

**For researchers:** A coalition focused on collective bargaining for information could provide a venue for researchers who study data value estimation or data bargaining to make broader impact on the informational commons. Researchers working directly on core technical questions of data scaling, data ablation, cryptography, information theory, and related topics could see increased impact for their work. This coalition could also provide opportunities for researchers across disciplines to collaborate on examining the AI industry and to inform policymakers working towards healthy data markets. In general, we expect that the formation of a formal or informal coalition for CBI would serve to greatly increase the impact of data-centric ML/AI research.

**For information producers:** This coalition could provide support for information producers immediately concerned about revenue sustainability and labor organizers who might be facing near-term negotiations in which AI and information economics constitute the central issues. The coalition could facilitate negotiation with tech companies for terms, conditions, and compensation desired by information producers.

**For the tech industry:** The existence of a coalition for *CBI* could also offer support for safety efforts at AI companies and labs that seek to mitigate potential harms of their systems. As laid out in Section 3, by working with data coalitions more closely and scaling data carefully, AI companies will have more granular oversight of their systems and be more capable of predicting potential harms. Moreover, engaging with collective bargaining will also improve data provenance and authenticity, and thereby help to build trust among their consumer bases.

**For people who just want more capable (safe) AI:** Our argument for *CBI* is not just about avoiding harms from AI. Collective bargaining can actually lead to better AI systems in the short, medium, and long term. A data marketplace that maintains its own foundations can create richer and more accurate records, and therefore better models. People provide higher quality data when they are aligned with the project of collecting and using the data because they have a stake in the gains. Without this alignment, people will stop producing information in good faith, and/or focus on keeping important information secret and resisting surveillance, and we will be stuck in a world in which models are built only on what systems are able to scrape/glean/surveil, not on people's pro-socially motivated information production.

## 1.4 Alternative Views

There are several possible alternatives to *CBI* that could prevent market failures and foster sustainable AI advances, but each has its own limitations compared to the impact on information markets of *CBI*, if implemented.

Often the first approach mentioned in policy, governance, and safety conversations is for state actors to regulate AI from the top down. If regulators are able to formulate enough laws that say "AI labs cannot use AI for some task" or "AI labs must complete economic risk assessment and pay some kind of data-dependence tax", perhaps certain bad outcomes could be avoided. This might involve either arm's-length regulation or direct public ownership or management of AI.

However, our view is that traditional regulatory approaches cannot address all of the issues we outline, in light of the political capital wielded by AI labs and other organizations that may stand to benefit from capital concentration, as well as more general political challenges. Regulation will also be limited in responsiveness and coverage, and it will be challenging to coordinate regulation across sovereigns. And it seems unlikely that any regulatory approach that does not feature support for *CBI* can reproduce incentives for the production of local information goods critical to intra-sovereign communities and geographies.

Another view to consider is that leaning heavily into open AI can act as an effective approach for power decentralization, obviating the need for *CBI*. This approach may fail to fully mitigate the risks laid out above insofar as (1) the best-performing "frontier" models retain or expand a performance advantage versus open source performance levels, and/or (2) the ability to use or profit from open models depends on access to capital [57].

## 2 Issues with Markets for Information

This section details the challenges and issues posed by AI to information markets.

### 2.1 Competitive Markets Do Not Efficiently Provision Capital to Informational Goods

Actors without market power, who exert effort to produce information goods that other market actors value, struggle to receive compensation. Such failures diminish incentives to create new informational goods.[2]

These properties affect AI policy, training practices, and competition between AI labs. For example, AI labs train on scraped and sometimes pirated content [27]; then other labs seek to train on the output of leading models [58]. Just as the non-excludability of information harms the interests of

---

[2]Other related problems include well-known issues like free-riding [22] and adverse selection due to a buyer's inability to appraise information (the classic "market for lemons" [5]). While our primary focus here is on organizations seeking to be paid for information production, e.g., news, science, and writing, sellers of information in other settings also stand to benefit from our proposed solution.

actors who produce or invest in primary information goods such as research or creative work, it also impacts the economics of AI model development. This has led to increased research attention at the intersection of AI and copyright [21, 45].

Designs, ideas, and other non-IP-protected information goods have always been easily copied, deterring investment to an unknowable degree. Intellectual property regimes (e.g., patents) attempt to remedy this by creating temporary monopolies on certain informational patterns. But in doing so, they also create constraints on society-wide exploitation of certain valuable information. There is no consensus as to the optimal balance between the incentives to create information and the freedom to exploit it [18, 8, 46]. But in practice, information goods are more often produced by actors who already have market power – or good prospects of acquiring it – because this gives them the ability to translate information into further competitive advantage or profit.

Our argument relies on the fact that AI progress transforms information processing, making the exploitation of information significantly more efficient. By examining how this change is likely to impact incentives for information creation, we can use insights from Arrow [7] and others to reason about the potential for AI-driven power concentration.

## 2.2   AI Progress and Changing Incentives around Information

Because competitive markets tend to drive the value of information goods down to their marginal cost, close to zero, it is instead *informational frictions* and *non-market factors* that create incentives to produce information goods. Important informational frictions include (1) intellectual property rights, i.e., artificial monopolies and (2) *de facto* barriers to informational processing, i.e., the difficulty of extracting complex and non-obvious insights from information. Important non-market factors include: (1) barriers to informational access, i.e., secrecy, (2) intervention by non-market actors, i.e., government investment in knowledge production, and (3) private market power, i.e., monopolies enabling exclusive exploitation of information.

We believe AI progress, broadly construed, will tend to reduce the economic importance of informational frictions. In general, AI will make it easier for all actors to locate and extract value from publicly available information, including the non-protectable aspects of otherwise-protectable IP. Barring unlikely expansive interpretations of IP rights, this shift widens the gulf between the social value of information and the return that information producers are able to earn.

Consequently, AI will increase the importance of non-market factors in sustaining incentives to produce information. Information good producers may elect to rely more on secrecy to exploit a greater share of the value of their information, and/or reduce their investments. This would correspondingly increase the need for government investment to generate socially optimal levels of open information production. These dynamics will also increase the returns to actors enjoying market power that puts them in a unique position to exploit information. Already in the pre-AI era, monopolists or near-monopolists within particular industries were important investors in information production biased toward their spheres of exclusive exploitation. AI will tend to increase their incentives both to fund production of information goods that benefit their interests, and also to keep that information secret, diminishing the "accidental" public benefits of private R&D.

A further consequence is that the reduction of informational friction increases the economic importance of capital. In a world where many actors have access to technology that enables them to fully analyze the informational commons, the relevant "friction" becomes access to capital. Many actors will have similar and expansive knowledge about how to exploit the information in the commons; but benefits of that knowledge will flow more efficiently to those most able to act on it, namely those with most access to capital.

Several likely outcomes are foreseeable: we list some (in non-exhaustive fashion) below:

**Labor substitution and automation**: Most simply, as AI gains new capabilities, AI systems will decrease the economic value of many human tasks [3]. This is particularly likely with human tasks that involve extracting complex insights from public information (such extraction comprises a significant aspect of academic research); and/or leveraging knowledge which is rare among humans but not fully unique to individuals, such as sharing and applying specialized training.

**Dampened incentives for creative content creation:** The current AI paradigm threatens the viability of copyright protection (see e.g. [21, 25]). Even if training is not fair use, and *a fortiori*

if it is, AI reduces the proportion of information's total social value that generates a return to information producers. This makes the incentives for producers of expressive work look more like the incentives for open source software developers: their primary motivation may become merely altruistic, or limited to personal reputational effects (a factor in contributions to Stack Overflow or GitHub [37]). However, making matters worse, AI may also complicate or reverse the incentives of altruistic non-market agents who invest in information production. If it becomes apparent that public information strengthens private capital or misaligned political actors more than it serves the public good, benevolent governments and individuals will cease to contribute to informational commons such as open science and free software.

**Increased dependence on government intervention**: Information goods oriented towards the general cultural good or commons will depend increasingly on patronage by non-market actors such as governments. This dependence would pose threats to the sustainability of the commons.

**Entrenchment of private market power**: More speculatively, firms in a unique position to exploit information may end up performing an increasing share of investment in informational goods. This will bias knowledge production toward information goods oriented toward the narrow interests of particular firms, not the general interests of the population.

**Potential for winner-take-all dynamics or zero-sum manipulation**: Relatedly, financial returns to AI that derive from individual or population-level manipulation (such as advertising or political influence) may tend to be winner-take-all. Larger, more powerful AI systems that can efficiently cause many individuals to take actions that benefit the owners of these systems can out-compete weaker systems with less-manipulative or more public-interest goals, exacerbating the difficulty of raising capital to support the latter types of systems.

## 2.3   AI-driven Power Concentration and a "Capital Singularity"

The points raised above will have implications for fine-grained policy interventions, business strategy, and understanding the macroeconomics of AI more generally [3, 24]. However, our focus in this paper is on a particular class of extreme outcomes: unprecedented power concentration in large pools of capital, that we might call a capital singularity.[3]

Capital consolidation means (a) larger pools of capital, and/or (b) more concentrated decisional power steering capital pools, e.g., greater authority delegation to a decisive executive or agent. AI has the potential to increase both non-linearly.

Historically, there have been open empirical questions around the relationship between capital consolidation and capital returns [48] (i.e., When do smaller hedge funds outperform larger ones? When do larger technology companies outcompete smaller ones?). The answers to these questions have always depended on context; but we argue that, in light of the arguments above, AI progress is likely to tilt key contextual factors toward advantaging consolidated capital. In a world where many actors have access to technology that enables them to comprehensively analyze the informational commons, the relevant frictions or barriers to exploiting such non-excludable information become not intelligence, fresh ideas, or a unique interpretive point of view (as was often the case in the past decades) but rather capacity to *act quickly and/or forcefully* on analyses that are ultimately available to everyone. Thus, even if many actors have similarly expansive access to a vast and evolving knowledge commons, the benefits of that knowledge will flow efficiently to those most able to act on it, which is likely tightly correlated with access to capital.

Critically, this opens the door to an extreme feedback loop. The market could eventually discover that the clearest path to financial returns in the AI economy is to pool capital as much as possible and delegate decision power to a few highly-empowered (human or AI) agents. The brakes on this process of market-driven power concentration will reflect not a diversity of market needs and strategies, but rather hard, non-market-based firewalls: informational barriers (secrecy) and political fissures (capital controls).

---

[3]Capital singularity is different from how the term "singularity" is used in AI safety discussions and draws attention to the concentration of power and capital.

# 3 Harms from AI-driven Power Concentration

Many scholars have written at length about AI-driven power concentration, including political economy [50, 29] and AI safety [30] researchers. Power concentration also has implications for ongoing discussions about catastrophic risk more broadly construed [23]. Having laid out why the current trajectory of the AI industry is likely to intensify information market failure and power concentration, we now expand on why this is undesirable.

**Political and moral issues:** Massive concentration of the returns to information risks major political disruption and moral harms. AI's economic disruption has been previously compared to the Industrial Revolution [2, 14]. Even without any unemployment, mass under-employment across sectors could lead to political instability. Even if labor disruption is slower than some have predicted (or is successfully mitigated by unforeseen developments), there is also serious concern that AI-driven power concentration could threaten governance integrity by disrupting social ties or democratic processes.

Finally, capital consolidation could impact qualitative aspects of human moral and political life. Creativity and social roles could be homogenized and devalued as models begin to transform them. So too could morality and politics, as more AI systems mediate the processes by which people acquire moral and social norms.

**Market failure harming information ecosystems and technical progress**: If a few dominant AI operators control a disproportionate share of the economic value of all information, this could paradoxically lead to AI systems that are *less capable* than the systems in a counterfactual world with more diverse information markets, and stronger incentives for "information-producing ecosystems". Moreover, if the few leading AI systems would not properly incentivize information production from the general public, the overall pool of "fresh" training data would shrink [36] (and the pool of engaged *training data contributors* may also become smaller), then it could be the case that AI gets worse along some dimensions or improves more slowly than it otherwise might. In a worst scenario, it is possible for there to be a collapse in both supply and demand for truthful, human-produced information, after which information markets would traffic only in manipulative or low quality information.

**Everything in public creates opportunities for power concentration**: Without collective bargaining, information ends up in one of two states: (1) under private control of an agent, or (2) out in public. It has been argued by open data advocates that moving information from state 1 to state 2 mitigates power concentration. However, all information in state 2 empowers agents subject to a coefficient representing their ability to exploit it. As we have argued, this coefficient is probably tightly coupled to capital. This means that if all information is out in public, we may differentially empower hyper-powerful, hyper-decisive, and hyper-self-interested actors. Such actors may be less attentive to general political and moral concerns, and therefore more likely to precipitate safety disasters.

*CBI* gives us a third state, namely information held by trusted representatives of large classes of information producers, and who may then titrate information, with caution, to self-interested third-parties. These trusted representatives are not aiming simply to maximize power or wealth; they have broader, more complex concerns like diversity in information, the well-being of their set of stakeholders, and/or the maintenance and development of a particular aspect of culture. Because they are empowered to represent larger aggregated pools of data, they enjoy bargaining power that individual agents do not. By mitigating the influence of more narrowly self-interested agents within groups, they maximize the good for groups as a whole. By increasing potential returns on investment in information goods, they preserve incentives for knowledge production.

**Risks of unpredicted harms and unchecked control**: If the current trend of improving AI by scaling up computing power persists, AI model structures will become more opaque and as a result, predicting harms will be increasingly challenging. Moreover, as the selected few gain more computational power, they will gain nearly exclusive control over AI models.

*CBI* points to an alternative path to scaling computation–scaling data, and the data-centric AI approach [35, 55] will foster more conscious, informed decisions about identifying training data and provide some "ballast" against extreme or exploitative plans. Moreover, *CBI* enables a feedback loop between data intermediary organizations and AI companies, and in doing so, *CBI* bakes some amount of accountability into AI.

**Capital Singularity as a Safety Issue**: The capital singularity scenario would precipitate extreme safety risks because any economically-dominant actor – whether human or AI – would need to set aside safety concerns and social accountability in order to achieve its pole position. In short, an adverse selection process favoring unsafe actors would occur in the consolidation of capital.

Consequently, we believe *CBI* is a tractable strategy for addressing important catastrophic risks. If AI progress creates such risks inherently, and cannot be practically stopped or slowed in a general sense, then mitigating the adverse selection of powerful actors in an AI-mediated political economy may be the best available risk-reduction measure.

**Lack of privately held information as a safety issue:** Understanding the risks and harms of extremely powerful AI systems requires testing and auditing with privately held information that was previously not available to the AI system. To assess or remediate AI safety, non-market actors such as governments, researchers, data intermediaries, and information "guilds" must leverage their non-public information to evaluate the performance and risks of AI, similar to how studies of algorithmic biases were carried out. If all or most economically-significant information is held by economically-motivated actors, these actors will seek to merge into each other, eliminating the possibility of leveraging privately held information to gauge AI safety.

## 4   Roadmap towards a CBI Coalition

We argue that it will be critical to societal health to support *CBI* with a broad definition of information producer. As more tasks and industries are exposed to a booming AI capabilities manifold [17], more people will find themselves dependent upon AI in some capacity. To this end, we should create an inclusive big-tent coalition that supports collective bargaining around data work, data labor, creative work, intellectual property production, knowledge work, skilled labor, and more, leveraging existing social structures such as online communities, unions, non-profits, firms, and sectoral coalitions.

In general, our expectation is that coordinating large-scale collective action (ideally involving co-ordination between major unions, major players in content creation domains like journalism and academia, and new emerging grassroots organizations) will lead to the most desired outcomes.[4]

In our introduction, we led with concrete actions that could be pursued by information-producing industries (e.g. science, journalism), regulators (e.g. the FTC and DOJ), the AI safety community,

---

[4]For the purposes of considering all extreme outcomes, it is worth noting that collective action from labor could create "labor cartel" conditions which are also unhealthy for markets. However, we think the current balance of power is such that this is not a serious concern in the near-term. Still, it is useful to consider that in some cases CBI may slow down some domain-specific progress in the short term, but (we believe) to overall large benefits in both AI safety and AI capabilities.

Table 1: Agenda for collective data bargaining and safer AI adoption

| Short name | Stakeholder | Recommendation |
|---|---|---|
| **Data Joint Ventures** | **Information–producing industries & communities** | Form joint ventures that pool data and negotiate licensing terms; block un-negotiated AI training routes. |
| **Safe Harbor** | **Regulators (FTC, DOJ, etc.)** | Publish guidance or rules stating that such collective-bargaining ventures do not violate antitrust law. |
| **Interp & Attrib** | **ML research community** | Advance interpretability and valuation methods (influence, attribution) that quantify data's causal impact. |
| **Human-Data UX** | **HCI & Design communities** | Create human-data interaction paradigms that give individuals transparent control over how their data is used. |
| **Support from AI Safety** | **AI safety advocates** | Direct funding and advocacy toward the collective-bargaining agenda. |
| **Support from Tech** | **Large technology companies and established AI labs** | Embrace *CBI* for trust building, harm mitigation, and a "pay your customers" principle. |

ML research, HCI research, and tech companies. We restate them here in Table 1. Below, we further discuss how these actions complement each other.

*CBI* currently faces legal, technical, and social challenges to materialize. Legal concerns may in fact be the most foundational (and hence feature prominently in our suggestions): collective bargaining arrangements at a large scale – despite actually mitigating a stark power imbalance between AI builders and information producers – may be perceived as anticompetitive, and if so, information producers may hesitate to organize due to fears of antitrust consequences. As such, policymakers should prioritize expanding permissible coordination on the information production side, or clarifying CBI's permissibility under antitrust laws.

As *CBI*-related joint ventures and intermediaries gain legal clarity, technical and social problems can be tackled in parallel. On the technical side, *CBI* requires reliable data management systems that enable escrow, transparent valuation, and provenance tracking [11, 34, 39]. Information producers will need access to some reasonable estimate of data value so that they can reason about whether any given bargaining proposition is a "good deal". This should include both machine learning research on data value estimation (**Interp & Attrib**) and human-computer interaction contributions (**Human-Data UX**) such as new interfaces and paradigms for people to interact with and control the flow of data (e.g., new browser extensions, platforms, and AI literacy interventions). Concretely, almost all technical advances in data-centric AI, including advances in understanding valuation (at the observation or group level), scaling laws, poisoning, and adversarial attacks, can help to reduce information asymmetries around the value of data and improve CBI outcomes.

In terms of social challenges to *CBI*, we believe the broad coalition can be most successful if we (1) leverage "natural" sectoral coalitions (e.g., medicine, journalism, academia) (2) leverage national-level public interest organizations (e.g. state-supported data trusts), and also (3) support the creation of trusted data intermediaries around new, uncollected data like neural link data, high-resolution video, detailed genetic and health tracking data, etc. Combining these different types of data intermediaries will foster a healthier overall ecosystem. Intermediaries might also contribute to a data commons, perhaps using Creative Commons licenses with an exception only for market leaders, similar to the EU's very-large-platform definition.

To summarize, we believe that collective bargaining around the use of information for AI will be a critical and necessary intervention to prevent massive power concentration and associated harms to society. We laid out a number of arguments for why this power concentration is likely in the absence of such collective bargaining. We believe that healthier information markets can be achieved, but this will require coordinated action on legal, technical, and social fronts. This action will have the greatest chance of success if supported by a broad coalition of information producers and academic, civic, and industry organizations.

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
