# OpenReview forum: "Collective Bargaining in the Information Economy Can Address AI-Driven Power Concentration"
_NeurIPS.cc/2025/Position_Paper_Track — NeurIPS 2025 Position Paper Track_

### Official Review · Reviewer_5AVj · 2025-07-28

[review text omitted: it was posted to a different submission]

---

### Official Review · Reviewer_49Nr · 2025-08-05

**Significance:** 3
**Presentation:** 3
**Rating:** 5
**Confidence:** 3

**Summary:**

This position paper argues that collective bargaining is a vital solution to address power imbalances in AI-driven information markets. It criticizes how large AI companies take advantage of individual data contributors and suggests collective bargaining to give data producers more power when it comes to negotiating data use, pay, and limits. Using economic theory, historical examples, and new research, the paper shows the dangers of capital consolidation and calls for fair data governance to help the AI ecosystem as a whole.

**Strengths:**

This paper presents a compelling case for collective bargaining as a transformative approach to data governance in the era of AI. Its significance lies in its potential to reshape power dynamics and enhance the development of AI.

-Timeliness: Addresses pressing AI ethics and governance issues.
-Interdisciplinary: Integrates economics, law, and technology effectively.
-Practicality: Offers actionable steps for stakeholders, from regulators to tech companies.

**Weaknesses:**

-Limited Examples: Lacks concrete case studies to demonstrate CBI’s real-world impact.
-Counterarguments: Mentions open data advocacy but doesn’t fully refute it (Page 8, Lines 345-351).
-Implementation Details: Technical and legal challenges are identified but not deeply explored.

**Questions:**

Q1. Could you provide examples of successful collective bargaining from other fields that might apply to AI data markets? Like, how would you address technical challenges like data valuation and tracking in a collective bargaining system?

Q2.Might collective bargaining unintentionally create new power concentrations, such as data intermediaries, and how would you mitigate this?

**Alternative Position:**

Yes, and alternative positions are well-considered and named but not addressed

**Author Identification:**

No.

**Context:**

2

**Details Of Ethics Concerns:**

The paper explores collective bargaining as a mechanism to address power imbalances in AI and data governance, focusing on economic and structural issues. It does not involve human subjects, sensitive data, or raise significant concerns related to privacy, discrimination, safety, or environmental impact. The proposal aligns with ethical AI principles by promoting fairness and equity.

**Discussion:**

3

**Ethics:**

["NO or VERY MINOR ethics concerns only"]

**Position:**

Yes, the paper argues for or against a position related to machine learning.

**Support:**

3

**Thoroughness:**

4

---

### Official Review · Reviewer_HA5T · 2025-08-14

**Significance:** 3
**Presentation:** 3
**Rating:** 7
**Confidence:** 4

**Summary:**

This paper advocates that we should collectively support collective bargaining in the information economy in order to mitigate power concentration due to AI.

The specific proposal covers four points: First, industry and information producers should form ventures and groups, spanning the sector, to advocate on behalf of those who produce information. This includes journalists, researchers, creatives and others. Second, governments should relax antitrust, data protection and other rules with respect to multi-stakeholder entities that are advocating for the rights of information producers. Third, advocacy and research should be targeted so as to highlight the core contributions of data providers to the functioning of these models. And fourth, tech and industry players should support the agenda.

**Strengths:**

Unlike many of the other submissions I’ve read, this paper is unmistakably a position paper. It is argumentative, it takes a point that others could disagree with, it argues in a complete and comprehensive way, etc. I would recommend accept.

**Weaknesses:**

I don’t think you need to make claims about the capital “singularity,” which tends to make particular assumptions about functional forms and growth. I think you need weaker claims to make the point, and using this terminology, while flashy, undermines the argument.

Small nits:
“However, our view is that a fully regulatory approach faces challenges because of political capital wielded by AI labs and other organizations that think they stand to benefit from capital concentration, as well as more general political challenges” —> there’s a typo in here such that I couldn’t follow the sentence.

**Questions:**

When does state intervention or action to empower collective bargaining become state interference in undesirable form?

What specific institutional tools could be used to estimate the value of contributors' data in these collectives, given the AI model performance and other outcomes relevant to achieving equitable bargaining outcomes?

**Alternative Position:**

Yes, and alternative positions are well-considered and addressed by the argument

**Author Identification:**

No.

**Context:**

3

**Discussion:**

3

**Ethics:**

["NO or VERY MINOR ethics concerns only"]

**Position:**

Yes, the paper argues for or against a position related to machine learning.

**Support:**

3

**Thoroughness:**

3

---

### Note · Authors · 2025-09-03

**1-10 Additional Comments:**

We noticed that Reviewer 5AVj’s comments referred to data annotators specifically, whilst our definition of information producers is much broader and not limited to data annotators. They also raised some points that are out of the scope of our paper. We weren't sure exactly how to respond to this situation, so are flagging it here for the ACs.

**1-11 Submit Again:**

Probably yes

**1-1 Submission Process:**

4

**1-2 Next Year:**

It could be interesting to add sub-tracks.

**1-3 Future Development:**

It might also be interesting to experiment with an even larger number of small reviews for the position paper track, to get more eyes on more positions.

**1-4 Interest:**

["Panel discussions with other position paper authors", "Structured debates on controversial topics"]

**1-5 Thoughtful:**

8

**1-6 Supportive:**

8

**1-7 Technical Aspects Versus Position:**

7

**1-8 Gate Keeping:**

8

**1-9 Camera Ready Changes:**

Our reviewers have recommended several important clarifications and we plan to address them accordingly for a camera ready copy of our paper:
- Provide a list of concrete examples of organizations currently supporting Collective Bargaining for Information (CBI)-style interventions, such as Cloudflare, ProRata, OpenMined
- One reviewer provided valuable feedback on our use of the term “capital singularity”. We will better distinguish our discussion from other uses of “singularity” in AI safety and AI ethics. We aim to talk specifically about concentration of capital, and we will clarify this throughout our manuscript, including through more careful phrasing and the use of more precise terms.
- We will aim to provide more concrete recommendations for the ML research community in particular.
- One reviewer raised a question about whether CBI would lead to unintended concentration of power. We will dedicate space to discuss concerns with “overcorrection”and why we are not worried about this in the short term.
- We will aim to further clarify how our work fits into existing research and labor initiatives on cooperatives, gig worker organizing, etc.
Please share any additional comments, suggestions, or feedback about your experience with the position paper track.

**3-1 Review Response1:**

H5AT

**3-2 Reaction To Review1:**

We deeply appreciate this reviewer's feedback on the use of “singularity”. As we note above, a focus in our minor edits to a camera ready will be to add more clarity about our focus on concentration of power in capital-holding organizations vs. the use of singularity in AI safety more generally. We will clarify our specific claims (and ultimately follow the advice to avoid flashiness or over-stating, while retaining a version of our core concern, which is a positive feedback loop of power accumulation).

We also appreciate the suggestions to further discuss undesirable levels of state interference and more specific institutional tools for value estimation.
- Re: determining an undesirable level of state interference, our general approach is to advocate for a rights-based approach in the short term – giving content organizations certain rights to organize without concern for anti-trust (as heuristic, equal rights for coordination of data sellers as for data buyers) and giving individuals necessary rights of data control such that they can participate in collective bargaining.
- Re: institutional tools for data valuation, we will further clarify the role of technical data attribution and value estimation. (echoes response to 49nr)

**3-3 Review Response2:**

49Nr

**3-4 Reaction To Review2:**

We appreciate the suggestion to further ground our position paper in current concrete examples. One focus of our minor edits for a camera ready version of the manuscript will be to directly include a short list and discussion of real-world example organizations (e.g.: recent efforts from Cloudflare, the startup ProRata.ai, the non-profit OpenMined).

Re: tools for data valuation, we will further clarify the role of technical data attribution and value estimation. (echoes response to H5AT)

**3-5 Review Response3:**

5AVj

**3-6 Reaction To Review3:**

One focus of our revisions will be to further connect our suggestions to existing thinking in labor economics (and relevant critical fields) and existing work in organizing gig work, creating new cooperative models, etc.

We also highlight that a goal of our paper is not to provide new qual or quant data about specific workers or data annotators, but rather to provide a broad position about the role of collective bargaining in a post-AI age. We define information producers as people who produce valuable information in digital formats, including but not limited to crowdworkers, knowledge workers, writers, creative professionals.

---

### Meta-Review · Area_Chair_zB5v · 2025-09-09

**Rating:** 7
**Confidence:** 3

**Strengths:**

The following strengths were highlighted in the reviews:

- The position of the paper is clear and the paper provides a compelling arguments for it.
- The position is timely and actionable.
- The paper offers an interdisciplinary perspective on the position.

All the reviewers positively evaluated this works and its contributions.

**Weaknesses:**

The following concerns were highlighted in the reviews:

- The paper doesn't provide a great number of examples to support CBI, and the counterarguments are not extensively discussed. Additionally, practical aspects, including technical and legal challenges, could have been more deeply explored.
- The paper could be strengthened by incorporating interdisciplinary and global perspectives and analyzing demographic and socio-economic differences among workers.
- Some of the claims in the paper may be too strong, and the terminology used may undermine the arguments.

The authors promised to addressed the reviewers' concerns in the camera-ready version of the paper, and are encouraged to do so.

**Questions:**

The reviewers asked multiple interesting questions; please refer to the questions in the reviews below. I would particularly emphasize the questions related to the practical aspects, since these, in my opinion, were not extensively discussed in the authors' response.

**Ethics:**

No apparent ethical concerns.

**Thoroughness:**

3

---

### Decision · Program_Chairs · 2025-09-26

Accept